# The experience of pedagogical training on postgraduate rehabilitation health professionals: A qualitative study

Gianluca Bertoni[1,2,3], Benedetto Giardulli[1], Barbara Minozzi[2,4], Ilaria Coppola[5], Laura Furri[6], Marco Testa[1], Simone Battista[7] *

**1** Department of Neurosciences, Rehabilitation, Ophthalmology, Genetics, Maternal and Child Health, University of Genoa, Savona, Italy, **2** Department of Clinical and Experimental Sciences, University of Brescia, Brescia, Italy, **3** Training Unit, Azienda Socio-Sociosanitaria Territoriale di Cremona, Cremona, Italy, **4** Training Unit, Azienda Socio-Sociosanitaria Territoriale di Mantova, Mantova, Italy, **5** Department of Education Sciences, School of Social Sciences, University of Genova, Genova, Italy, **6** School of Medicine and Surgery, University of Verona, Verona, Italy, **7** School of Health and Society, Centre for Human Movement and Rehabilitation, University of Salford, Salford, Greater Manchester, United Kingdom

* s.battista@salford.ac.uk

**Data Availability Statement:** All relevant data are within the manuscript and its Supporting Information files.

## Abstract

### Introduction

Health professionals that act as lecturers in higher education necessitate the acquisition of pedagogical skills along with clinical ones. Consequently, pedagogical training courses have been introduced as part of professional development or within university curricula. While several studies explored the experiences of attending courses on pedagogical methodology, there is a notable gap in the literature within the rehabilitation field. Hence, this qualitative study explored the experience of Italian postgraduate health professionals in rehabilitation about the experience of attending pedagogical methodology courses in their postgraduate education.

### Materials and methods

A qualitative focus group study was carried out. Specifically, the authors purposefully recruited participants with different professional backgrounds (physiotherapy, speech therapy, and others) with or without lecturing experience. Recent graduates and students of the Master of Science in 'Rehabilitative Sciences of the Health Professions' at the University of Verona (Verona, Italy) were recruited. The focus groups were analysed following a 'Reflexive Thematic Analysis' by Braun & Clarke within a social constructionist framework.

### Results

Three focus groups were conducted with seventeen Italian participants (age: 33 ± 9; 71% women, n = 12; 29% men, n = 5). The analysis identified three main themes: 1) "A Brave New Pedagogical World," reflecting participants' exposure to innovative teaching approaches; 2) "Becoming a Cutting-Edge Lecturer," highlighting skills acquired for

**Funding:** The author(s) received no specific funding for this work.

**Competing interests:** The authors have declared that no competing interests exist.

delivering inspiring lectures; and 3) "Something Beyond Pedagogy," where participants reported skills applicable to other professional contexts, including clinical practice.

## Conclusion

The results of this study showed that pedagogical courses provide a positive learning experience for rehabilitation health professionals, helping them develop relevant pedagogical skills. Although our findings suggest the potential benefits of these courses in preparing healthcare professionals for teaching roles, further studies are needed to evaluate their direct impact on educational practices and patient outcomes.

## Introduction

Health professionals who act as lecturers in higher education necessitate acquiring pedagogical skills to enhance the efficacy of their lectures [1]. Being a clinical expert does not necessarily mean being an expert educator. Most educators in the medical field need to receive adequate training in pedagogy [1]. Without specific training, they resorted to appealing to their personal experience, using those patterns they experienced as students through a 'modelling' process and relying on their cultural sensitivity and capacity for elaboration [2]. As healthcare services evolve and become more complex, it is essential for educators to go beyond such 'modelling' processes [3]. Graduates in health professions are now expected to possess clinical expertise and engage in evidence-based practices, following patient-centred care [4]. Educators may struggle adequately preparing students for these multidimensional demands without structured pedagogical training. Therefore, configuring a well-defined pedagogical training programme for new lecturers is crucial to support high-quality graduate outcomes in both clinical and educational practices [1].

To answer this educational need, different European countries adopted various strategies to improve teaching qualities to answer that need. For instance, they established 'Teaching and Learning Centres' to provide lecturers with faculty development programmes to promote and structure lecture training activities [5]. These centres comprise faculty developers, instructional designers, and educational technologists [5]. Various European countries (e.g., the United Kingdom, Sweden, Norway, Finland, Germany and Dutch) have developed specific requirements or guidelines for university teacher training (e.g., postgraduate certificate in education) [6]. Lecturers' professional development was found to be positively linked to improved students' learning experience with an encouraging impact on performance-based metrics [7]. In Italy, structured pedagogical training for healthcare professionals is less standardised than in other European countries [8]. While some universities offer pedagogical courses within specific health programmes, a national framework or requirement for pedagogical certification is not yet established or needed to teach in higher education. This creates variability in teaching quality across institutions [9]. The Master of Science (MSc) in 'Rehabilitative Sciences of the Health Professions' is an example of a postgraduate programme aiming to bridge this gap. Established with a unique curriculum that includes courses in pedagogical methodology, curriculum design, and tutorial methodology, this MSc programme attracts a diverse range of students who obtained a Bachelor of Science (BSc) in one of the eight rehabilitation health professions recognised by the Italian health professional register [10–12].

Different qualitative and quantitative studies have explored the experience of attending courses in pedagogy [6]. In general, participants who attended pedagogy-focused courses

reported a positive impact of these courses across various domains of their lives [13], including improvements in lecturing skills such as listening and communication, professional benefits such as career advancement, and personal gains such as fostering healthier relationships with family, patients, and colleagues [13]. However, these studies rarely focused on health professionals working in rehabilitation (e.g., physiotherapists) and from geographical areas other than the Northern European countries. These considerations are noteworthy as the meanings attached to education are influenced by ethnicity, living area, and working background [14–16]. Moreover, they mainly focused on faculty development programmes attended those who were already lecturing and as highlighted elsewhere, "those who need faculty development the most, attend the least" [17]. In light of the above, the purpose of this qualitative study was to explore the experience of Italian postgraduate health professionals in rehabilitation (with or without previous teaching experiences) in taking pedagogical courses.

## Materials and methods

### Study design

A qualitative focus group study within a social constructionist framework was adopted to answer the research question. Focus groups are technical tools for investigating a research question through participants' discussion [18]. The study complied with the Declaration of Helsinki, ensuring ethical standards were upheld to protect the rights and welfare of participants, and was reported according to the Consolidated Criteria for Reporting Qualitative Research (COREQ) [19]. Ethical approval was obtained from the Approval Committee for Human Research (CARP), University of Verona, Verona, Italy (Approval date: 18/05/2023; Code: 31/2022).

### Participants

Recent graduates and students of the MSc in 'Rehabilitative Sciences of the Health Professions at the University of Verona (Verona, Italy) were recruited. This MSc focused on research methods and methodologies, teaching in higher education and health economics. This MSc programme focuses on research methods, pedagogy in higher education, and health economics. It includes a unique curriculum with mandatory courses such as pedagogical methodologies, curriculum design, and tutorial methodology. The programme attracts a diverse cohort of students with a BSc in one of the eight rehabilitation health professions recognised by the Italian health professional register, including physiotherapists, speech therapists, occupational therapists, orthoptists, podiatrists, psychiatric rehabilitation technicians, psychomotor therapists, and professional educators. All courses within the degree programme's curriculum are mandatory, and the complete course curriculum can be viewed in S1 Table.

Participants were purposely sampled to identify those who were more informative to answer the research question by ensuring that the group was as heterogeneous as possible [20]. To achieve this, we sought participants with relevant experience in rehabilitation, education, and management. Specifically, the authors contacted the course leader of this MSc, asking her to reach out to the former or current students of the programme based on different criteria. The course leader contacted the students via email. She sent an email outlining the aim of the study, its design and the informed consent to be signed. She specifically explained in the email to contact BM instead of her to programme the focus groups. This decision was made to prevent the course leader from knowing which students participated in the focus groups, as she taught a course related to pedagogical methodologies. Participants were recruited based on their professional backgrounds, such as physiotherapy, speech therapy, and others, to ensure a balanced representation of various rehabilitation areas. Additionally, we included

professionals with varied levels of lecturing experience (or none), those with roles in coordinating staff and departmental operations, and clinical tutors who work closely with students. Individuals with expertise in coordinating staff and departmental operations alongside clinical tutors were specifically included to capture insights on the course's impact from professionals accustomed to managing team dynamics and mentoring students in clinical settings. Their perspective was deemed valuable for assessing the development of pedagogical and interpersonal skills relevant to educational and clinical practice. This approach aimed to capture a broad lens of experiences related to the pedagogical courses. Additional information regarding the ethical, cultural, and scientific considerations specific to inclusivity in global research is included in S1 File.

The focus groups were structured to increase variability inside them. The sample size was guided by the concept of 'information power' [11] rather than the commonly used but methodologically inappropriate for RTA 'data saturation' [12]. According to the information power model, the number of participants needed depends on factors such as the study aim, sample specificity, theoretical background, quality of dialogue, and analysis strategy [11]. Given the expertise of the researchers in qualitative research and pedagogy, the solid theoretical foundations of our study, the specificity of our research question, and the purposeful selection process, an estimate of 16 to 18 participants divided into three focus groups was considered, including at least one individual with clinical expertise, one with lecturing experience, and one with management/coordination experience per each FGD. Furthermore, the authors aimed to diversify the group by considering participants' gender, age, and years of enrolment in the MSc.

## Data collection method

A semi-structured focus group guide was constructed by BM IC and SB (Table 1). BM is a physiotherapist with experience in teaching and tutoring. SB is a physiotherapist, PhD in 'Neurosciences' and 'Medical Sciences' and temporary lecturer in 'Pedagogical Methodology for Health Professionals'. IC is a psychologist with a PhD in 'Migrations and intercultural processes'. SB and IC are trained in qualitative methodologies, with expertise in conducting qualitative studies, and have provided BM with all the necessary training to conduct and analyse

**Table 1. Semi structured focus group guide.**

| QUESTIONS | DOMAINS |
|---|---|
| Thinking about the topics covered during the pedagogical courses, which expectations did you hold and which were not met? | Introductory question expectations and motivations |
| Thinking about the topics covered in these courses, which elements struck you the most (either positively or negatively)? | Strengths and weaknesses |
| Describe your experience as a student in these courses in three words. For each word used, can you give an explanation? | Reflection on experience |
| Before participating in these courses, what was your perception of the lecturer? Did the topics covered in the lessons change your perception of the role of the lecturer? | Reflection on experience |
| In what way do you think you could use the skills obtained during the courses? Can you give me some examples? | Acquired Skills |
| Are there any topics addressed in these lessons that may connect with your field of work? | Acquired Skills |
| Thinking about the relationship with the patient or the team, how can what you have learnt to be helpful? Can you give me some examples | Acquired Skills |
| If you had to describe this course to a colleague, what would you tell them? | Closing question |
| *Would you like to add something that was not mentioned?* | Wrap-Up |

Focus Groups Discussions (FGD), following her during this process. BM and IC identify as women; SB identifies as a man.

The 'dual moderator' focus group included eight open-ended questions to investigate the participants' experience of pedagogical courses [21]. One moderator (IC), with expertise in qualitative research, ensured that the focus group adhered to the correct procedures, while the other (BM) followed an interview guide to ensure all contents were addressed. Together, the moderators facilitated the progression of the focus group interview, ensuring that all topics were thoroughly investigated [21]. The initial outline was tested for relevance and comprehensibility of content in two pilot interviews conducted online with two former MSc students. Following these interviews, the semi-structured focus group guide was revised and corrected to the final version. The three FGD were conducted online by BM and IC in September 2023. In each FGD, participants were informed of the recording of the session, reassured that the data would remain anonymous, that they were free to answer or not answer questions, and that they could leave the session at any time. Prior to each FGD, the contents and topics covered within pedagogical courses in the MSc study's objective were summarised. After the researchers introduced the study's objectives and summarised the topics covered in the pedagogical courses, each participant briefly introduced themselves to the group. The FGD opened with an introductory question concerning expected and unfulfilled expectations regarding the courses. Then, moderators asked them which topics they found most relevant, positively or negatively. Afterwards, moderators asked them to describe their experience in these courses focusing on the figure of the lecturer. The authors tried to analyse any change of perspective regarding the lecturers' role after the experience and the topics covered in these courses. Then, the focus shifted to the skills gained from these courses, either for teaching or other fields (e.g., clinical). The FGD ended by asking the participants whether they would recommend (or not) these courses to other colleagues and why. At the end of the group interview, space was left for further consideration. The FGD were recorded and then transcribed *verbatim*. The data and recordings were stored in a database on an online university drive. The recordings were deleted as soon as the transcription phases were completed. The transcription of the FGD was anonymised and shared with the other researchers.

The FGDs were conducted in Italian, the native language of all participants, to ensure that their experiences and perspectives were captured authentically. This approach allowed the researchers to generate themes directly from participants' words in their mother tongue, thus preserving the integrity of their responses during the analysis phase. After themes were established, selected quotes and themes were translated into English for this manuscript. The translation was performed by the research team members fluent in Italian and English to ensure that the translated content remained as faithful as possible to the participants' original meanings.

## Data analysis

In addition to BM, the data analysis involved three other researchers, GB and BG. They are physiotherapists and PhD students in 'Neurosciences' with expertise in conducting qualitative studies. GB and BG identify themselves as men. SB guided the authors throughout the data analysis. As a first step, a descriptive sample analysis was carried out to identify data regarding age, gender, professional category and previous experience in clinical, teaching, tutoring and coordination, as well as the academic year they attended the MSc.

Qualitative data were analysed using Braun and Clark's 'Reflexive Thematic Analysis' (RTA) to identify the patterns of meaning and consequently generate themes relating to the experiences of the pedagogical courses [22]. RTA was selected for analysing the qualitative data due to its flexibility and adaptability, which allowed us to capture the nuanced

**Table 2. Six steps of the RTA.**

| Phases | Process | Authors' Involvement | Authors' Actions |
|---|---|---|---|
| 1) Data familiarisation | All authors immerged themselves in the data to understand depth and breadth of the content. | All authors became familiar with the data in this phase. This process is fundamental to contacting the data and taking notes of impressions and insights. | • Reading and re-reading data set;<br>• Listening to the audio recordings;<br>• Taking notes;<br>• Marking transcripts sections relevant to the research question. |
| 2) Coding | Three authors started to generate codes to organise the dataset, giving full and equal attention to all data items. | BM, GB and BG systematically coded the data. They adopted semantic data coding. | • Peer debriefing: memos were shared during research meetings for reflexive thoughts;<br>• Labelling and organising data items into meaningful groups. |
| 3) Generating initial themes | Three authors started to generate initial themes by sorting codes and identifying meaning of and relationships between codes. | BM, GB and BG generated initial themes separately, clustering similar codes. | • Diagramming or mapping to make sense of theme connections;<br>• Writing themes and their defining properties. |
| 4) Reviewing and refining themes | Authors reviewed the initial themes, reworking or eliminating some until finding a set of themes fitting the dataset. | All authors reviewed the coding and initial themes separately and then jointly generated three themes that fit the most data. BM, GB, BG and SB reviewed the agreed themes against the codes and the entire dataset. | • Ensuring there is enough data to support a theme;<br>• Re-working and refining codes and themes. |
| 5) Defining and naming themes | Authors developed themes names and refined them as they could tell a 'story'. | All authors finalised the final themes and their definitions. Themes were reviewed to ensure the themes represented participants' experiences and perspectives. | • Peer debriefing and team consensus on themes;<br>• Cycling the data and the identified themes in order to organise the story. |
| 6) Producing the report | All authors produced the final report and refined the themes if necessary. | BM selected the illustrative quotations from the focus group, and all authors reviewed and agreed. SB led the writing of the paper, and all authors participated in this phase. | • Writing the final report;<br>• Report on reasons for theoretical, methodological, and analytical choices. |

perspectives of participants in pedagogical training courses. RTA is particularly suitable for studies with an experiential focus, as it enables a deep exploration of participants' personal insights and the meanings they attach to these experiences. This approach allowed us to generate themes inductively, ensuring that they authentically reflected participants' voices without imposing pre-existing frameworks. Moreover, RTA facilitated a collaborative analysis process, enriching the rigor of theme development and aligning with the social constructionist framework of this study. From the point of view of the epistemological framework, this study adopted a social constructionist approach and an experiential orientation. The coding was mainly at a descriptive/semantic level. To analyse the focus group data, we followed the six steps of RTA as outlined by Braun and Clarke, as shown in Table 2.

## Results

Seventeen Italian participants (age: 33 ± 9; 71% women, n = 12; 29% men, n = 5) took part in the FGD (Table 3). Each FGD lasted about 90 minutes. Three themes were developed from the analysis of the FGD. These themes underscored participants' experience attending courses in pedagogical methods, focussing on perceived knowledge and skills gained from these courses: 1. 'A Brave New Pedagogical World'; 2. 'Becoming a Cutting-Edge Lecturer'; 3. 'Something Beyond Pedagogy'. The following sections describe each theme in detail, with tables in S2 Table that highlight the primary quotes leading to the generation of each specific theme.

### Theme 1: A brave new pedagogical world

This theme revolves around the transformative experience participants described as encountering a "brave new pedagogical world," shifting their understanding of teaching from a

**Table 3. Demographic of participants.**

| ID | Age | Gender | Professional role |
|---|---|---|---|
| *Focus Group 1* | | | |
| P1 | 26 | Woman | Speech therapist with teaching experience |
| P2 | 48 | Man | Physiotherapist in a coordinating role |
| P3 | 28 | Woman | Physiotherapist with tutoring experience |
| P4 | 29 | Man | Physiotherapist with clinical experience |
| P5 | 31 | Man | Physiotherapist with clinical experience |
| *Focus Group 2* | | | |
| P6 | 34 | Man | Physiotherapist with teaching experience |
| P7 | 36 | Woman | Physiotherapist with teaching experience |
| P8 | 50 | Woman | Physiotherapist as teaching tutor on a degree course |
| P9 | 33 | Woman | Physiotherapist as teaching tutor on a degree course |
| P10 | 29 | Woman | Speech therapist with clinical experience |
| P11 | 34 | Woman | Physiotherapist with tutoring experience |
| *Focus Group 3* | | | |
| P12 | 26 | Woman | Speech therapist with clinical experience |
| P13 | 25 | Woman | Speech therapist with clinical experience |
| P14 | 52 | Man | Physiotherapist in a coordinating role with teaching experience |
| P15 | 27 | Woman | Physiotherapist with teaching experience |
| P16 | 28 | Woman | Speech therapist with teaching experience |
| P17 | 27 | Woman | Psychiatric rehabilitation therapist with clinical experience |

ID, participants unique code

technical skill to a relational and dynamic practice. The researchers identified several codes that helped construct this theme (Table 1 in S2 Table). A key aspect of this theme is *the importance of good communication in pedagogy*, where participants began to recognise that effective teaching requires expertise and the ability to communicate clearly and engage students. One participant noted, "A good teacher isn't just a guru with knowledge, but someone who can communicate effectively" (P7, 36, physiotherapist with teaching experience). This reflection shows a shift from viewing teachers as merely content experts to seeing them as facilitators who actively shape the learning environment. Another essential element is *the importance of taking care of students*, illustrated by how participants felt valued and supported by their instructors. One participant shared that this caring approach was meaningful: "I felt important to the teachers; they made me feel comfortable and provided all the essential materials" (P6, 34, physiotherapist with teaching experience). Experiencing this level of care firsthand helped participants understand the impact of a student-centred approach, reinforcing the importance of empathy and attentiveness in pedagogy. Participants also highlighted *the importance of a consistent and good model* in teaching, where the alignment between teachers' actions and their pedagogical principles served as an inspiring example. One participant observed, "The remarkable coherence between what Professor X explained and how they taught was impressive," adding that this consistency created a "wow" moment that left a lasting impression (P10, 29, speech therapist with clinical experience). This experience emphasised how leading by example in teaching methodology could profoundly influence students' perceptions of teaching. The theme further includes *the importance of playing an active role in learning*. Rather than absorbing information passively, participants were encouraged to be active creators, directly engaging with the material through hands-on tasks and collaborative projects. As one participant described, "The active learning methodologies we used opened up a whole new

world for me" (P10, 29, speech therapist with clinical experience). This code reflects a shift toward understanding learning as a participatory process, deepening participants' appreciation for methods that invite students to actively engage and take ownership of their learning. Participants also expressed *an awareness of the behind-the-scenes effort in teaching* as they came to realize the extensive planning, organisation, and commitment required to create effective lessons. One participant reflected, "I realised how challenging and complicated it can be to build a lecture" (P15, 27, physiotherapist with teaching experience). This insight fostered a new-found respect for teaching as a complex and labour-intensive discipline, going beyond what they initially expected. Finally, the theme encompasses *discovering how pedagogy can be fascinating*, as some participants reported developing a genuine interest in teaching. For instance, one participant noted, "I never imagined that teaching could be so fascinating. I started with no expectations, but by the end, I felt an unexpected interest in becoming a lecturer" (P15, 27, physiotherapist with teaching experience). Collectively, these reflections illustrate how the experience opened up new perspectives for participants, expanding their view of teaching as not just a skill but a meaningful and inspiring profession.

### Theme 2: Becoming a cutting-edge lecturer

This theme reflects participants' desire to become "cutting-edge lecturers" who integrate technical expertise with dynamic, student-centred teaching strategies to enhance learning outcomes. The researchers identified several codes that contributed to constructing this theme (Table 2 in S2 Table). A foundational code within this theme is *knowledge of adult learning mechanisms*, which many participants found transformative. Participants expressed how understanding the distinct learning needs of adults allowed them to adapt their teaching styles more effectively. One participant noted, "The course showed us how adults learn and the importance of meeting different learning styles" (P3, 28, physiotherapist with tutoring experience), emphasising that this insight opened new possibilities for crafting lessons that resonate with adult learners.

Another significant code, *how to tailor lectures to students*, illustrates participants' growing commitment to making lessons relevant to students' real-world experiences. Participants highlighted how they adjusted their lectures to integrate practical examples, thereby connecting theoretical content with clinical applications. As one participant shared, "I tried to modify my lessons, bringing in concrete examples from clinical practice" (P8, 50, physiotherapist and university tutor), which helped them reframe their role as facilitators who guide students in applying knowledge to practical settings. *How to evaluate students properly* emerged as another key component, where participants embraced assessment frameworks such as Bloom's taxonomy to foster critical thinking and clinical reasoning. One participant described the changes in their assessment approach, stating, "I redesigned my exams to encourage clinical reasoning, and I use a learning contract to set clear expectations" (P8, 50, physiotherapist and university tutor). This approach aimed to promote transparency and fairness, contributing to a more equitable and supportive learning environment. Participants also emphasised *how to use active learning tools* to create interactive and engaging experiences. They described experimenting with methods like group projects, videos, and individual tasks to make lectures more participatory and impactful. "I used group projects, videos, and individual tasks to make lectures interactive," one participant explained, reflecting on the benefits of these strategies in making learning more dynamic and student-centred (P9, 33, physiotherapist and university tutor). *How to communicate effectively with students* was identified as an essential skill in capturing and maintaining student attention. Participants discussed how adopting effective communication techniques, such as modulating tone, using movement, and incorporating digital tools,

enhanced classroom engagement. One participant emphasized, "It's not just what you teach but how you convey it. Using tone, movement, and digital tools helped me keep students engaged" (P9, 33, physiotherapist and university tutor), underscoring the importance of adaptability and presence in fostering a vibrant learning environment. Finally, *how to arrange pedagogical tools and project effective lessons* captures participants' increased proficiency in structuring and planning educational content. They learned to design lessons thoughtfully, ensuring that each component was aligned with learning objectives and tailored to students' needs. One participant remarked on this approach: "Every time I have to present, I think about making the presentation captivating, with images and elements to keep attention high" (P14, 52, physiotherapist and coordinator with teaching experience). This planning allowed participants to envision themselves as forward-thinking educators, ready to inspire and connect with students in meaningful ways. Together, these codes capture the essence of becoming a "cutting-edge lecturer" by focusing on preparation, adaptability, and a strong commitment to student-centered teaching.

### Theme 3: Something beyond pedagogy

This theme explores how the pedagogical skills participants gained extended beyond the classroom, significantly enriching their clinical practice and personal lives. The researchers identified several codes that contributed to constructing this theme (Table 3 in S2 Table). The first code, *teamwork skills*, captures how participants developed a stronger appreciation for interdisciplinary teamwork through collaborative projects. This experience enhanced their ability to communicate and engage constructively with colleagues from diverse backgrounds. As one participant reflected, "The group work allowed us to learn from each other and work together toward a common goal" (P16, 28, speech therapist with teaching experience). This collaborative experience strengthened their professional relationships and built a foundation for effective teamwork in various settings. Another core code is *how to learn from experience*, highlighting the participants' newfound appreciation for self-assessment and growth through reflection. This reflective mindset helped participants approach new challenges with openness and adaptability, as one noted: "The courses encouraged me to reflect on my actions and ask questions, which helps me approach new challenges with an open mind" (P11, 34, physiotherapist with tutoring experience). This emphasis on reflection was seen as critical for both professional development and personal growth. *Developing a new awareness that enhances an open mindset* emerged as an important outcome, as participants expressed a renewed willingness to embrace change and challenge themselves. One participant described this transformation, stating, "The course has inspired a desire to continue, to learn, and to change" (P2, 48, physiotherapist in a coordinating role). This mindset shift was accompanied by *an increased desire to do things well*, with participants describing how the course sparked a motivation to pursue excellence in their work. One participant explained, "I came out of the class wanting to do better; it gave me the desire to strive for quality" (P3, 28, physiotherapist with tutoring experience). This drive was seen as foundational to their ongoing personal and professional development. The code *enhancing willingness to change for good* further underscores how the training inspired participants to embrace continuous improvement. This growth-oriented perspective encouraged them to view challenges as opportunities for personal and professional advancement, fostering resilience and adaptability in their careers. *Becoming a critical thinker* was another key code, where participants learned to approach training and educational methods with a discerning eye. This critical perspective empowered them to evaluate teaching methods more effectively and prioritize those most beneficial to learners. "These courses have taught me to look critically at teaching methods and select what truly benefits students," explained one

participant, underscoring how this skill would support their future development (P1, 26, speech therapist with teaching experience). *Applying communication skills in clinical, personal, and professional settings* was also transformative for participants. They reported that improved communication made their interactions with patients, colleagues, and family members clearer and more impactful. One participant shared, "I've applied what I learned in didactics to better communicate with patients and colleagues, making my interactions clearer and more impactful" (P6, 34, physiotherapist with teaching experience). This adaptability helped participants build rapport and understanding across diverse interactions. Lastly, *increasing patient adherence based on their learning mechanisms* reflects how participants transferred their pedagogical insights into clinical practice. Recognising that tailoring communication to patients' individual learning styles could boost engagement, one participant remarked, "Knowing how to adapt my approach based on learning styles has improved my interactions with patients, especially in motivating them to follow through on exercises" (P7, 36, physiotherapist with teaching experience). This application of pedagogical principles to patient care underscores the relevance of these teaching skills in clinical settings, ultimately enriching participants' ability to connect with and motivate their patients. Collectively, these codes reveal how the skills gained in pedagogical training extended beyond the classroom, enabling participants to enhance their effectiveness in clinical practice, strengthen their personal and professional relationships, and adopt a mindset of continuous learning and adaptability.

## Discussion

The present study aimed to investigate the experience of pedagogical courses to a group of health professionals in the rehabilitation field with or without a previous background in pedagogy. In light of the above, the analysis of the FGD led to the identification of a sort of common pathway taken by the student that starts by experiencing an entirely new world of conceiving and applying pedagogy (Theme 1. A Brave New Pedagogical World). Through this experience, the students understood the strategies to become inspiring and cutting-edge lecturers (Theme 2. Becoming a cutting-edge lecturer). Finally, their learning experience went beyond pedagogy, as they learnt different skills they could apply in fields other than pedagogy, such as clinical and personal ones (Theme 3. Something Beyond Pedagogy).

From the data analysis, most participants started the courses on pedagogical training without specific expectations. However, actively experimenting the illustrated pedagogy methodologies revealed the practical value of using communication and tools designed to capture and maintain students' attention and interest. This hands-on approach highlighted the importance and benefits of structuring pedagogy around 'active learning' principles. As a result, this experience fostered their motivation and interest in a new way of delivering education (Theme 1: A Brave New Pedagogical World). Motivation and perceived usefulness are fundamental elements contributing to adult learning and developing new skills [23].

From the data analysis, it is evident that the general perception of lecturers and how to deliver education have been transformed. The lecturer who dispenses their knowledge has given way to the communicative and motivating lecturer who places the student at the centre, building and guiding them in their learning process. This kind of lecturer is who the participants want to be inspired by and want to become: a facilitator of learning, motivators, leaders, and performers [4]. In this new way of lecturing, the students felt comfortable, supported, and able to achieve their goals, and this is how they want their students and collaborators to feel. According to Schumacher et al., motivation is a powerful consideration for optimising learning [23].

The participants also discovered how complicated lecturing can be, how many skills are needed, how much energy and time it requires, and how fascinating and vital it is to achieve excellent results. This discovery instilled in them a strong desire to teach, making lecturing personally relevant for them. Personal relevance is a powerful instrument to motivate students and energise learning [24]. Many participants who had never fathomed the idea of teaching as a professional outlet hoped to become a lecturer at the end of these courses. Therefore, the participants increased their awareness of who they wanted to become and what to do to get there. Thus, they began the growth process that will lead them to acquire new skills (Theme 2. Became a cutting-edge lecturer).

Participants who were already lecturing reported they were already applying what they learned in their lectures, obtaining excellent feedback from their students. They learnt pedagogical competencies related to the diverse learning modalities of adults. Thanks to these competencies, participants learned how to create appropriate learning contexts, facilitating their students through this process. Then, data analysis outlined the perceived acquisition of methodological and teaching competencies related to applying innovative teaching methodologies and active learning, which are known to improve students' learning process [25]. Afterwards, the participants felt they improved and learned organisational and management competencies in pedagogy, allowing them to collaborate in developing continuous professional development courses and designing a module based on specific contexts' needs. Participants felt they learnt communication competencies to adequately transmit lesson content while empathising with their students, inspiring their learning process. Communication is the dominant factor affecting the academic achievements of students. The assessment of students' perceptions of the role of teachers' communication skills in their academic success indicated that effective teaching not only depends upon the knowledge base of the teacher but also on the style of teacher communication skills [26].

Moreover, participants reported being able to provide adequate feedback and fair examinations. Regarding feedback, they perceived they understood how to use it to approach and entertain students to stimulate their attention and participation. Effective feedback reinforces good practice, motivating learners towards desired outcomes [27]. Yet, students often criticise feedback as infrequent and inadequate [27]. As per the examination, the participants felt they could carry out an appropriate and objective assessment with particular attention to the learners' experience and course objectives. This approach aligns with the so-called 'constructive alignment', an integrative design for lecturing in which the consistency between learning outcomes, learning activities and assessments is emphasised to guide students' learning [28].

Finally, thanks to these courses, participants demonstrated they developed skills that can be transferred to contexts other than the education field, such as the workplace and social and personal contexts (Theme 3. Something Beyond Pedagogy). From the data analysis, participants reported that group works during the pedagogical courses contributed to developing collaborative learning among them, appreciating and understanding the importance of learning with others by constructing meanings, asking questions and collaborating. Meeting in interdisciplinary groups brings several benefits; these include sharing ideas, developing community, reducing feelings of being alone, receiving feedback, sharing commonalities across disciplines, learning from diverse experiences, and creating supportive connections [29]. These courses allowed the participants to engage and collaborate with various health professionals through collaborative learning. This learning method has determined the development of teamwork skills they transferred to their workplace, improving their ability to relate to other professionals. Working efficiently in a team while creating a good atmosphere is fundamental to improving the quality of patients' care and the workplace's well-being [30]. Specifically for rehabilitation, teamwork and a multidisciplinary approach to care are paramount to achieving

therapeutic success because of the complexity and rising prevalence of chronic care conditions [31]. Moreover, the transformative experience reported by participants aligns with adult learning theories such as the "Transformative Learning Theory" and the "Experiential Learning Theory" [32, 33]. "Transformative Learning Theory" emphasises 'critical reflection' to reconsider pre-existing beliefs and attitudes about one's profession aiming at taking informed action and implementing new knowledge [32]. "Experiential Learning Theory", rooted in constructivist philosophy, highlights the role of interpersonal interactions and contextual learning as fundamental to experiential learning [33]. This approach enables learners to translate, or refine, learning in new contexts to meet new challenges [33]. For instance, participants reported shifting what they learnt on pedagogy into their clinical practice.

The experience of the pedagogical courses was seen by the participants as a source of personal and professional growth, developing their reflective abilities, encouraging them to ask questions, making connections between different knowledge, re-evaluating their work critically, and setting the ability to learn from their experiences. These abilities are reflected in participants with increased levels of awareness, decision-making, and professional responsibility, which are indispensable qualities for health professionals in the rehabilitation field [34]. Then, the participants experienced a renewed desire for dynamism, learning, putting themselves to the test, and constantly challenging themselves, highlighting the development of an open-mind. In a growth-oriented mindset, challenges and mistakes are seen as opportunities [35]. These challenges led to greater resilience in the workplace, and a greater predisposition to change, which led the participants to seek new opportunities and new job prospects and to rethink their way of being and valuable work skills in this modern era characterised by transformation and uncertainty [36]. From the analysis of FGD, the participants reported the skills they learned had a positive impact on their clinical practice, especially in their relationship with the people they were caring for. The key to successful rehabilitation therapy also lies in the relationship established between the health professional and the individual, and communication is a fundamental aspect of this [37]. Therefore, courses focussing on effective communication can improve patient's quality of care. By applying what they learned, the participants realised they could be more straightforward and more impactful in conveying information and providing feedback, perceiving better communication between them and the patients.

Applying the learned knowledge about learning mechanisms within rehabilitation treatment allows for creating a personalised learning context with more significant rehabilitation proposals, which contribute to motivating patients and increasing their participation, ultimately determining the success of the treatment [38]. Patient education about managing their conditions is fundamental as different pathological conditions are chronic, and people must learn how to manage or live with them. Therefore, applying learning knowledge in this education process might positively impact people's outcomes. Motivation and participation in treatment are positive prognostic factors linked to the success of the treatment, and being skilled motivators is an essential competency for health professionals in rehabilitation. Finally, the communication skills learnt also positively impact personal and social relationships, allowing for better relationships with colleagues and family members.

Different limits of this study need to be acknowledged. First, some of the participants' considerations might stem from other courses attended during their MSc and not solely from their participation in the pedagogical courses. However, the authors specified that the questions in FGD wanted to stress their present experience. Additionally, the authors analysed the experience of an MSc whose pedagogical courses can be different from others taught elsewhere. Nevertheless, the courses' topics and objectives followed syllabi updated with the latest evidence on pedagogy. Then, only a few rehabilitation healthcare professionals (i.e., physiotherapists, speech therapists and psychiatric rehabilitation technicians) were reached with a

majority of physiotherapists. Moreover, most of them were white women. Finally, all the interviewees lived in a similar geographical area (i.e., northern Italy). These considerations are essential to highlight since meanings attached to education might be influenced by gender, ethnicity and area of living. That notwithstanding, it is fundamental to hear the voices of these students as they have the potential to shed some light upon ways to improve education in Mediterranean countries. However, future studies should consider other ways to account for more diverse participants.

## Conclusions

This study highlighted the importance and value of incorporating pedagogical courses into a postgraduate programme for health professionals in rehabilitation due to the extreme usability and transferability of the knowledge and skills to which they give access. Our findings highlight the potential benefits of pedagogical programmes focused on active learning, interdisciplinary collaboration, and communication skills for healthcare professionals in rehabilitation. These findings lay a foundation for future research on investigating the experience of pedagogical programmes in other countries to form updated curricula for rehabilitation healthcare professionals. Participants gained specific pedagogical expertise by participating in these courses to deliver captivating lectures and inspire their students. Moreover, these courses provided a platform for acquiring skills that transcend pedagogical skills and improved the work and personal contexts of the participants. These results aligned with previous evidence focused on other realities, expanding the knowledge of the positive experience of attending pedagogical courses. To conclude, pedagogical courses can unlock unexpected potential in the rehabilitation field, enhancing both educational practices and broader professional competencies. By implementing these courses, educational institutions and healthcare organisations might effectively prepare healthcare professionals to excel in their teaching roles while improving patient care and overall healthcare outcomes.

## Supporting information

**S1 Table. The curriculum of the Master of Science in 'Rehabilitative Sciences of the Health Professions' at the University of Verona.**
(PDF)

**S2 Table. Codes and themes.**
(PDF)

**S1 File. Inclusivity in global research.**
(PDF)

## Acknowledgments

We wish to extend our sincere gratitude to all participants of this study. Your invaluable contributions and willingness to dedicate your time to this research have significantly enriched our findings. This work would not have been possible without your involvement

## Author Contributions

**Conceptualization:** Gianluca Bertoni, Benedetto Giardulli, Barbara Minozzi, Ilaria Coppola, Laura Furri, Marco Testa, Simone Battista.

**Data curation:** Gianluca Bertoni, Benedetto Giardulli, Barbara Minozzi, Ilaria Coppola, Marco Testa, Simone Battista.

**Formal analysis:** Gianluca Bertoni, Benedetto Giardulli, Barbara Minozzi, Simone Battista.

**Investigation:** Gianluca Bertoni, Benedetto Giardulli, Barbara Minozzi, Ilaria Coppola, Simone Battista.

**Methodology:** Gianluca Bertoni, Benedetto Giardulli, Barbara Minozzi, Ilaria Coppola, Marco Testa, Simone Battista.

**Project administration:** Simone Battista.

**Supervision:** Laura Furri, Marco Testa, Simone Battista.

**Validation:** Benedetto Giardulli.

**Writing – original draft:** Gianluca Bertoni, Benedetto Giardulli, Barbara Minozzi, Ilaria Coppola, Simone Battista.

**Writing – review & editing:** Gianluca Bertoni, Benedetto Giardulli, Barbara Minozzi, Ilaria Coppola, Laura Furri, Marco Testa, Simone Battista.

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
