## [Decision Letter · Decision Letter 0]

25 Oct 2024

PONE-D-24-29648The Experience of Pedagogical Training on Postgraduate Health Professionals in Rehabilitation: a Qualitative StudyPLOS ONE

Dear Dr.  Battista,

Thank you for submitting your manuscript to PLOS ONE. After careful consideration, we feel that it has merit but does not fully meet PLOS ONE’s publication criteria as it currently stands. Therefore, we invite you to submit a revised version of the manuscript that addresses the points raised during the review process.

We look forward to receiving your revised manuscript.

Kind regards,

Mc Rollyn Daquiado Vallespin

Academic Editor

PLOS ONE

Journal requirements: When submitting your revision, we need you to address these additional requirements. 1. Please ensure that your manuscript meets PLOS ONE's style requirements, including those for file naming. The PLOS ONE style templates can be found at https://journals.plos.org/plosone/s/file?id=wjVg/PLOSOne_formatting_sample_main_body.pdf and https://journals.plos.org/plosone/s/file?id=ba62/PLOSOne_formatting_sample_title_authors_affiliations.pdf 2. Please include a complete copy of PLOS’ questionnaire on inclusivity in global research in your revised manuscript. Our policy for research in this area aims to improve transparency in the reporting of research performed outside of researchers’ own country or community. The policy applies to researchers who have travelled to a different country to conduct research, research with Indigenous populations or their lands, and research on cultural artefacts. The questionnaire can also be requested at the journal’s discretion for any other submissions, even if these conditions are not met.  Please find more information on the policy and a link to download a blank copy of the questionnaire here: https://journals.plos.org/plosone/s/best-practices-in-research-reporting. Please upload a completed version of your questionnaire as Supporting Information when you resubmit your manuscript. 3. Your ethics statement should only appear in the Methods section of your manuscript. If your ethics statement is written in any section besides the Methods, please delete it from any other section.  4. Please amend your list of authors on the manuscript to ensure that each author is linked to an affiliation. Authors’ affiliations should reflect the institution where the work was done (if authors moved subsequently, you can also list the new affiliation stating “current affiliation:….” as necessary). 5. Please match your authorship list in your manuscript file and in the system. 6. Please include captions for your Supporting Information files at the end of your manuscript, and update any in-text citations to match accordingly. Please see our Supporting Information guidelines for more information: http://journals.plos.org/plosone/s/supporting-information. 

Reviewers' comments:

Reviewer's Responses to Questions

**Comments to the Author**

1. Is the manuscript technically sound, and do the data support the conclusions?

Reviewer #1: Partly

Reviewer #2: Yes

Reviewer #3: Yes

2. Has the statistical analysis been performed appropriately and rigorously? 

Reviewer #1: N/A

Reviewer #2: Yes

Reviewer #3: Yes

3. Have the authors made all data underlying the findings in their manuscript fully available?

Reviewer #1: Yes

Reviewer #2: Yes

Reviewer #3: Yes

4. Is the manuscript presented in an intelligible fashion and written in standard English?

Reviewer #1: No

Reviewer #2: Yes

Reviewer #3: Yes

5. Review Comments to the Author

Reviewer #1: General reflection:

This manuscript has very important evidence that provide evidence on the impact of such a kind of trainings on the quality of practices in service provision and education. To make sure your knowledge reach to the global and local community who can benefit form this evidence, I strongly recommend you consider the revision recommended to improve your result and discussion presentations. Finally, it will be great to have a language support to improve the presentation. Thank you!

Abstract

Line #35+

Result: Participants description and the finding could be improved with better presentation as the current one is confusing. E.g. …with average age 33 ± 9 and majority women (n=12). Three themes were developed: 1) “A Brave New Pedagogical World” …. 2) ….. and 3) ….

Line #40+ Discussion: I recommend replacing ‘discussion’ by ‘conclusion’ as we are not actually discussing at the abstract. Also, this part states a kind of recommendation to implement this education. However, the study doesn’t test about the impact of this training on the impact of the training their education practice. Hence, we don’t have evidence to say it is effective and recommend for implementation at large scale. I think, there is another stage next to this study, evaluation of the impact of this training.

Introduction

Line #48-53: You set a foundation, but would make more sense to add a context where the existing education system is impacting the quality of graduates (in their service – be it in clinical or education practices) so as to justify the need to improve the pedagogical skills of educators could add value to the quality of graduates.

Line #60+: You could provide the Italian context of what is happening and what is know about the current practice.

Line #70: The statement “meanings attached to education are influenced by ethnicity, living area, and working background” require citation.

Line #78-79: The statement “The study complied with the Declaration of Helsinki” need be linked with the its purpose/function like the way you described after the connector “and”

Line #84+, Participants description: Some of the partis in the first paragraph could be moved to provide a context about Italy in the introduction section, but with citation.

Line #86-87: The sentences that started with “This MSc…” doesn’t flow well. Your want to indicate the structure and contents of the education program, but it needs improvement.

Line #96: The statement “purposely sampled to identify those who were more informative to answer the research question”, what criteria in your sampling process were used to identify those who are “more informative to answer the research question”? Some details about the criteria would add values.

Line #97-98: Same comment as above that is about “This MSc…”

Line #101-103: The point describe in this sentence is not convincing as relevant to the research questions as these roles are not specific to educational experience.

Line #125: use either the full term “focus group discussions” or FGD once you fully describe when you first use it.

Line #128: “At the beginning of each meeting …” seems that you have multiple meetings within a specific group. So, it is better to describe as “prior to each focus group discussions, ….”

Line #129: The statement “A brief presentation of each participant followed this introduction by the researchers” is confusing, so consider revision to add clarity or remove.

Line #122-139: Generally the interviewing process is too detailed, particularly since you have already included the interview guides, descriptions related to the flow of questions were a redundant point.

Line #143, Table 1: Consider improving the table format as the current structure is not in its best way.

Overall question on data acquisition: What was the language used to facilitate the focus group discussion (data generation process)? If you use any language other than English, what was it? When and how did you translate it to English language, providing that the manuscript is prepared in English? Who did the translation and why? I am forwarding these points as they are critical for the research rigor and for the quality of the evidence generated.

Line #146-150: I recommend to move most of these parts to the declaration of co-authors’ contribution.

Line #152-157, Data analysis: The reference guiding your analysis can be cited once at the beginning and then no need to repeatedly mention after each sentence in the same paragraph.

Line #156-157: The last statement here needs improvement for clarity, as it confuses due to citation of the table which you could have cited it at the end of the statement. Consider revision

Line #159, Table 2: I don’t think this content need a table as it is possible to describe the steps and actions taken at that stage, who did it, if needed.

Results

Line #164: use either the full term “focus group discussions” or FGD once you fully describe when you first use it.

Line #164-165: The statement “Three focus groups were conducted in September 2023 with seventeen Italian participants” is repetition and result is not its relevant place as you have already escribed it in the methods section.

Line #164-165: See my comment above, at the abstract section, the result part for a better way of presenting the data.

Line #165: Delete “Table 3 reported participants’ demographic information.” As t doesn’t add value, rather you may cite the table at some point in this paragraph.

Across the manuscript, search the words “focus groups” and try to either write them fully as “focus group discussions” or replace with its application (FGDs)

Line #169: The statement “the reader will find the description of each theme with …” need revision using active voice or the whole sentence need revision “In the following subsections, the reader will find the description of each theme with a table containing the primary quotes that brought the research to generate the specific themes.”

Across the manuscript, I would recommend to use Male/Female rather than Man/Woman.

Line #172, Table 3: Make sure the information about the working area couldn’t cause any ethical breaches as it can be used to relate with a specific participant by linking with the other characteristics across the document. Also, you may need to consider if this information is really relevant/would add value to the research anyways.

Line #173: Thee Legend “ID” may not directly identification number as it is not only a number and has a letter within the code. May be this can be “participants unique code”

Line #175-243: Presentation of the three themes

I can see there are three themes generated from the data and that you have presented them in detail within a table, with a number of quotes. That is a great depth, but that again makes the result to appear at very breadth level, and lack some level of depth analysis to present it as a manuscript. The paragraphs under each themes have great description and attempted to cover most of the points. However, non of them were structured as a conceptual description of the themes and sub-themes, if what you have presented in the table are sub-themes. It is highly recommended to describe them explicitly. Present each theme with their sub-themes and associated quotes need to be integrated with the presentation of your interpretation.

The tables have so much detailed quotes that can be used as additional file, if needed to provide the details, but the results need major revision to rewrite the themes and sub-themes with associated quote, probably one or two quotes could be presented, using various formats of presenting the quotes. I would strongly recommend the authors to refer other articles to learn how results are presented using quotes to support their argument/interpretation.

Discussion

Line $245+: The discussion section is well written, but authors attempted to discuss each piece of the information generated in the result section. Rather, I would recommend stepping back and see which points are most outstanding and deserve a focused discussion/argument to elevate the interpretation. This again will be linked to the earlier comment on the result section that require major revisions on its content. Therefore, I would recommend first to step back to the result section, complete the necessary revisions and come back to the discussion with the most powerful findings that you want to discuss with depth and made a conclusion on your study.

Acknowledgements

Line #367-368: I think you have participants who had invested their time and hence deserved to be acknowledged.

Thanks

Reviewer #2: The Article is well written and all the aspects are well explained. There sufficient information gathered to support the need and the significance of the the study. Methodology is well written to the extent of replication. The results of all the objectives have been met and illustrated. Discussion sound adequate and cover all the aspects.

Reviewer #3: Consider reorganizing the introduction to follow a more logical flow: starting with the general need for pedagogical skills among health professionals, followed by the existing strategies in different European countries, and then narrowing down to the specific gap concerning rehabilitation professionals in Italy.

It would be beneficial to highlight how your study's findings could contribute to the improvement of educational programs for health professionals, not just in Italy but potentially in other similar contexts.

When referring to 'Northern ones', consider using 'Northern European countries' or a more precise term to maintain clarity and consistency.

Some of the cited literature in the introduction section seems foundational but might be a bit dated. Including more recent studies or reviews on pedagogical training for health professionals, if available, could enhance the introduction's relevance.

How was the decision made regarding the number of focus groups (three groups) and the size of each (5-6 participants)? Was this based on a saturation point or other methodological considerations?

The study uses a 'dual moderator' focus group approach. Could you clarify why this particular method was chosen?

How were the online focus groups managed to ensure engagement and minimize potential distractions or technical issues? Were any strategies implemented to maintain the quality of interaction?

Why was Braun and Clark's Reflexive Thematic Analysis (RTA) selected for this study?

How did you determine the naming of each theme (e.g., 'A Brave New Pedagogical World')? Were these names derived directly from participants' language, or did they emerge from the researchers' interpretation of the data?

You mention the transformative experience that participants had regarding pedagogy. Consider expanding on how this experience aligns with adult learning theories (e.g., transformative learning theory or experiential learning theory). This can strengthen the theoretical underpinning of your findings.

6. PLOS authors have the option to publish the peer review history of their article (what does this mean?). If published, this will include your full peer review and any attached files.

Reviewer #1: No

Reviewer #2: **Yes: **Ahmed Ibrahim Al Kharusi

Reviewer #3: No

---

## [Author Response · Author response to Decision Letter 0]

17 Nov 2024

Authors’ general comment:

We would like to thank the Editor and the reviewers for their comments that improve the quality of our work. Please, find our rebuttal letter where we have highlighted in yellow all the amendments to our paper. 

Reviewer #1

Reviewer’s general comment:

Comments to the Author:

This manuscript has very important evidence that provide evidence on the impact of such a kind of trainings on the quality of practices in service provision and education. To make sure your knowledge reach to the global and local community who can benefit form this evidence, I strongly recommend you consider the revision recommended to improve your result and discussion presentations. Finally, it will be great to have a language support to improve the presentation. Thank you!

Authors’ reply: We thank the reviewer for their positive feedback on the importance of our study’s evidence. We have carefully addressed each of the reviewer’s suggestions.

Abstract

Comments to the Author:

Line #35+ Result: Participants description and the finding could be improved with better presentation as the current one is confusing. E.g. …with average age 33 ± 9 and majority women (n=12). Three themes were developed: 1) “A Brave New Pedagogical World” …. 2) ….. and 3) ….

Authors’ reply: We thank the reviewer for their valuable feedback. We have revised the Results section of the abstract to improve the clarity of the participant description and findings presentation. The participants’ demographics are now stated more clearly, and the key themes are presented in a simplified format to enhance readability

Authors’ action: 

Page 2, lines 35-39: Three focus groups were conducted with seventeen Italian participants (age: 33 ± 9; 71% women, n=12; 29% men, n=5). The analysis identified three main themes: 1) “A Brave New Pedagogical World,” reflecting participants' exposure to innovative teaching approaches; 2) “Becoming a Cutting-Edge Lecturer,” highlighting skills acquired for delivering inspiring lectures; and 3) “Something Beyond Pedagogy,” where participants reported skills applicable to other professional contexts, including clinical practice.

Line #40+ Discussion: I recommend replacing ‘discussion’ by ‘conclusion’ as we are not actually discussing at the abstract. Also, this part states a kind of recommendation to implement this education. However, the study doesn’t test about the impact of this training on the impact of the training their education practice. Hence, we don’t have evidence to say it is effective and recommend for implementation at large scale. I think, there is another stage next to this study, evaluation of the impact of this training.

Authors’ reply: We thank the reviewer for their insightful comments. We have revised the abstract to replace ‘Discussion’ with ‘Conclusion,’ as suggested. Additionally, we have adjusted the language in the conclusion to avoid suggesting large-scale implementation, emphasizing instead the potential implications of this training for further research.

Authors’ action: 

Page 2, lines 40-43: Conclusion: The results of this study showed that pedagogical courses provide a positive learning experience for rehabilitation health professionals, helping them develop relevant pedagogical skills. Although our findings suggest the potential benefits of these courses in preparing healthcare professionals for teaching roles, further studies are needed to evaluate their direct impact on educational practices and patient outcomes.

Introduction

Line #48-53: You set a foundation, but would make more sense to add a context where the existing education system is impacting the quality of graduates (in their service – be it in clinical or education practices) so as to justify the need to improve the pedagogical skills of educators could add value to the quality of graduates.

Authors’ reply: We thank the reviewer for this helpful suggestion. To provide a clearer justification for the need to enhance pedagogical skills among healthcare educators, we have expanded the Introduction to include context on how the current education system affects graduate quality, particularly in clinical and educational practices.

Authors’ action: 

Page 3, lines 51-56: Health professionals who act as lecturers in higher education necessitate the acquisition of pedagogical skills to enhance the efficacy of their lectures [1]. Being a clinical expert does not necessarily mean being an expert educator. Most educators in the medical field did not receive adequate training in pedagogy [1]. Without specific training, they resorted to appealing to their personal experience, using those patterns they experienced as students through a 'modelling' process and relying on their cultural sensitivity and capacity for elaboration [2]. As healthcare services evolve and become more complex, it is essential for educators to go beyond such ‘modelling’ processes [3]. Graduates in health professions are now expected to possess clinical expertise and engage in evidence-based practices, following patient-centred care [4]. Educators may struggle adequately preparing students for these multidimensional demands without structured pedagogical training. Therefore, configuring a well-defined pedagogical training programme for new lecturers is crucial to support high-quality graduate outcomes in both clinical and educational practices [1].

Line #60+: You could provide the Italian context of what is happening and what is know about the current practice.

Authors’ reply: We thank the reviewer for their suggestion. We have expanded this section to include the current context in Italy, describing the state of pedagogical training and any specific requirements or initiatives relevant to health professionals in higher education

Authors’ action: 

Page 3, lines 65-72: In Italy, structured pedagogical training for healthcare professionals is less standardised than other European countries [8]. While some universities offer pedagogical courses within specific health programmes, a national framework or requirement for pedagogical certification is not yet established or needed to teach in higher education. This creates variability in teaching quality across institutions [9]. The Master of Science (MSc) in 'Rehabilitative Sciences of the Health Professions' is an example of a postgraduate programme aiming to bridge this gap. Established with a unique curriculum that includes courses in pedagogical methodology, curriculum design, and tutorial methodology, this MSc programme attracts a diverse range of students who obtained a Bachelor of Science (BSc) in one of the eight rehabilitation health professions recognised by the Italian health professional register [10–12].

Line #70: The statement “meanings attached to education are influenced by ethnicity, living area, and working background” require citation.

Authors’ reply: Thank you for your constructive feedback regarding the statement that “meanings attached to education are influenced by ethnicity, living area, and working background.” I appreciate your attention to this matter. In response, I have added relevant citations to the manuscript that underscore how systemic factors such as race, ethnicity, socioeconomic status, and geographic location shape educational experiences and perceptions.

Line #78-79: The statement “The study complied with the Declaration of Helsinki” need be linked with the its purpose/function like the way you described after the connector “and”

Authors’ reply: Thank you for your insightful comment regarding the connection between the study's compliance with the Declaration of Helsinki and its ethical implications. I have revised the text to clarify that adherence to the Declaration was intended to ensure the protection of participants' rights and welfare throughout the research process. This emphasizes the importance of ethical standards in qualitative research. Thank you for your feedback, which has helped improve the clarity of this section.

Authors’ action: 

Page 4, lines 88-90: The study complied with the Declaration of Helsinki, ensuring ethical standards were upheld to protect the rights and welfare of participants, and was reported according to the Consolidated Criteria for Reporting Qualitative Research (COREQ).

Line #84+, Participants description: Some of the partis in the first paragraph could be moved to provide a context about Italy in the introduction section, but with citation.

Authors’ reply: We appreciate your insightful feedback regarding the participant description. In response, we have restructured the introduction to include relevant information about the context of Italy's pedagogical training for health professionals. This adjustment provides a clearer framework for understanding the significance of our study and the diverse backgrounds of the participants in the Master of Science program. We have ensured that citations are included to support the contextual information presented. This modification not only aligns with your suggestion but also enhances the overall coherence of the paper. Thank you for your valuable input.

Authors’ action: 

Page 3, lines 65-72: In Italy, structured pedagogical training for healthcare professionals is less standardised than in other European countries [8]. While some universities offer pedagogical courses within specific health programmes, a national framework or requirement for pedagogical certification is not yet established or needed to teach in higher education. This creates variability in teaching quality across institutions [9]. The Master of Science (MSc) in 'Rehabilitative Sciences of the Health Professions' is an example of a postgraduate programme aiming to bridge this gap. Established with a unique curriculum that includes courses in pedagogical methodology, curriculum design, and tutorial methodology, this MSc programme attracts a diverse range of students who obtained a Bachelor of Science (BSc) in one of the eight rehabilitation health professions recognised by the Italian health professional register [10–12].

Page 4, lines 97-102: Recent graduates and students of the MSc. in 'Rehabilitative Sciences of the Health Professions at the University of XXX (XXX, XXX) were recruited. This MSc focused on research methods and methodologies, teaching in higher education and health economics. This MSc programme focuses on research methods, pedagogy in higher education, and health economics. It includes a unique curriculum with mandatory courses such as pedagogical methodologies, curriculum design, and tutorial methodology. The programme attracts a diverse cohort of students with a BSc in one of the eight rehabilitation health professions recognised by the Italian health professional register, including physiotherapists, speech therapists, occupational therapists, orthoptists, podiatrists, psychiatric rehabilitation technicians, psychomotor therapists, and professional educators. All courses within the degree program's curriculum are mandatory, and the complete course curriculum can be viewed in the Supporting Information file 1.

Line #86-87: The sentences that started with “This MSc…” doesn’t flow well. Your want to indicate the structure and contents of the education program, but it needs improvement.

Authors’ reply: Thank you for your feedback regarding the clarity of the sentences about the MSc programme. We have already revised this sentence while addressing your previous comment to enhance its readability and overall flow.

Line #96: The statement “purposely sampled to identify those who were more informative to answer the research question”, what criteria in your sampling process were used to identify those who are “more informative to answer the research question”? Some details about the criteria would add values.

Authors’ reply: Thank you for your valuable feedback regarding the clarity of our sampling process. We appreciate your request for additional details about the criteria used to identify participants who were considered "more informative" for addressing the research question. In our revised manuscript, we have elaborated on the specific criteria employed during participant selection. We focused on individuals' relevant experiences in rehabilitation and education, which included their professional backgrounds (such as physiotherapy and speech therapy), as well as their levels of lecturing experience. Additionally, we considered participants’ roles in coordinating staff and departmental operations, as well as their experiences as clinical tutors, to ensure a comprehensive understanding of pedagogical courses.

These criteria were designed to capture a diverse range of perspectives, enhancing the richness of our qualitative data and ensuring that we could explore the educational experiences of health professionals from various rehabilitation disciplines.

Authors’ action: 

Page 4, lines 106-120: Participants were purposely sampled to identify those who were more informative to answer the research question by ensuring that the group was as heterogeneous as possible [20]. To achieve this, we sought participants with relevant experience in rehabilitation, education, and management. Specifically, the authors contacted the course leader of this MSc, asking her to reach out to the former or current students of the programme based on different criteria. The course leader contacted the students via email. She sent an email outlining the aim of the study, its design and the informed consent to be signed. She specifically explained in the email to contact BM instead of her to programme the focus groups. This decision was made to prevent the course leader from knowing which students participated in the focus groups, as she taught a course related to pedagogical methodologies. Participants were recruited based on their professional backgrounds, such as physiotherapy, speech therapy, and others, to ensure a balanced representation of various rehabilitation areas. Additionally, we included professionals with varied levels of lecturing experience (or none), those with roles in coordinating staff and departmental operations, and clinical tutors who work closely with students. Individuals with expertise coordinating staff and departmental operations alongside clinical tutors were specifically included to capture insights on the course's impact from professionals accustomed to managing team dynamics and mentoring students in clinical settings. Their perspective was deemed valuable for assessing the development of pedagogical and interpersonal skills relevant to educational and clinical practice. This approach aimed to capture a broad lens of experiences related to the pedagogical courses.

Line #97-98: Same comment as above that is about “This MSc…”

Authors’ reply: We have already revised this sentence while addressing your previous comment.

Line #101-103: The point describe in this sentence is not convincing as relevant to the research questions as these roles are not specific to educational experience.

Authors’ reply: Thank you for your comment regarding the inclusion of coordinators and clinical tutors to assess the course’s impact. We understand your observation that these roles may not be directly tied to formal educational experience; however, we believe their perspectives add valuable depth to our study. Their responsibilities in coordinating staff, managing departmental operations, and mentoring students involve frequent interaction with students and academic personnel, providing them with unique insights into how pedagogical training affects teaching and learning dynamics. As our analysis shows—particularly under the theme “Something Beyond Pedagogy”—these roles foster skills that are transferable to both educational and clinical settings, enhancing our understanding of the training’s broader impact.

To clarify this rationale further, we have revised the text to better highlight why we chose to include these professionals in our sample.

Authors’ action: 

Page 4, lines 116-120: Individuals with expertise coordinating staff and departmental operations alongside clinical tutors were specifically included to capture insights on the course's impact from professionals accustomed to managing team dy

---

## [Editor Report · Decision Letter 1]

19 Nov 2024

The Experience of Pedagogical Training on Postgraduate Rehabilitation Health Professionals: a Qualitative Study

PONE-D-24-29648R1

Dear Dr. BATTISTA,

We’re pleased to inform you that your manuscript has been judged scientifically suitable for publication and will be formally accepted for publication once it meets all outstanding technical requirements.

Kind regards,

Mc Rollyn Daquiado Vallespin

Academic Editor

PLOS ONE

---

## [Editor Report · Acceptance letter]

22 Nov 2024

PONE-D-24-29648R1 

PLOS ONE

Dear Dr. Battista, 

I'm pleased to inform you that your manuscript has been deemed suitable for publication in PLOS ONE. Congratulations! Your manuscript is now being handed over to our production team.

Kind regards, 

on behalf of

Dr. Mc Rollyn Daquiado Vallespin 

Academic Editor

PLOS ONE